# DecoMT: Decomposed Prompting for Machine Translation Between Related Languages using Large Language Models

**Ratish Puduppully**[1]   **Anoop Kunchukuttan**[3,5,6]    **Raj Dabre**[4]
**Ai Ti Aw**[1]    **Nancy F. Chen**[1,2]

[1]Institute for Infocomm Research (I[2]R), A[*]STAR, Singapore
[2]CNRS@CREATE, Singapore    [3]Microsoft, India
[4]National Institute of Information and Communications Technology
[5]IIT Madras    [6]AI4Bharat
puduppully@i2r.a-star.edu.sg   ankunchu@microsoft.com   raj.dabre@nict.go.jp
nfychen@i2r.a-star.edu.sg

## Abstract

This study investigates machine translation between related languages *i.e.,* languages within the same family that share linguistic characteristics such as word order and lexical similarity. Machine translation through few-shot prompting leverages a small set of translation pair examples to generate translations for test sentences. This procedure requires the model to learn how to generate translations while simultaneously ensuring that token ordering is maintained to produce a fluent and accurate translation. We propose that for related languages, the task of machine translation can be simplified by leveraging the monotonic alignment characteristic of such languages. We introduce DecoMT, a novel approach of few-shot prompting that decomposes the translation process into a sequence of word chunk translations. Through automatic and human evaluation conducted on multiple related language pairs across various language families, we demonstrate that our proposed approach of decomposed prompting surpasses multiple established few-shot baseline approaches. For example, DecoMT outperforms the strong few-shot prompting BLOOM model with an average improvement of 8 chrF++ scores across the examined languages.

## 1 Introduction

In this work, we focus on the translation between related languages, a vital aspect from both economic and social perspectives. A considerable amount of commercial activity and social interaction occur between neighboring regions speaking two related languages. In these situations, pivot translation via a third language, such as English, can prove inefficient due to two inference steps which can also cause cascading errors (Dabre et al., 2021). Instead, direct translation between related languages could significantly streamline trade and enhance social connections.

Related languages, often from the same family, share word order and lexical characteristics, leading to predominantly monotonic translations where word order is largely preserved. This is seen in languages like Hindi, Marathi, Malayalam, Tamil, Bengali, etc. from the Indian subcontinent, which follow a Subject-Object-Verb (SOV) structure. Similar monotonic translation relationships are also observed among other language pairs, such as Indonesian and Malay or Ukrainian and Russian.

Recent work has shown the power of few-shot prompting with large language models (LLMs) for tasks like machine translation, summarization, and question answering (Lin et al., 2022; Workshop et al., 2023). In machine translation, this approach prompts an LLM with a handful of example pairs and a test example. This requires the model to generate translations while ensuring a fluent word ordering, a process that fails to account for any unique characteristics intrinsic to the languages involved. For instance, it neglects the monotonic alignment—an integral trait evident in translations between related languages.

LLMs are often biased towards English in their training data. For example, in mT5 (Xue et al., 2021), Hindi and Malayalam tokens represent just 0.8% and 0.07% respectively. This imbalance hinders LLM performance in tasks involving non-English languages and English to non-English translations (Lin et al., 2022). In particular, for few-shot translation tasks between related languages, these models may not have encountered sufficient data in these languages. Overcoming these limitations can be achieved by incorporating inductive biases about related languages.

Recently, Khot et al. (2023) introduced an approach known as decomposed prompting. This

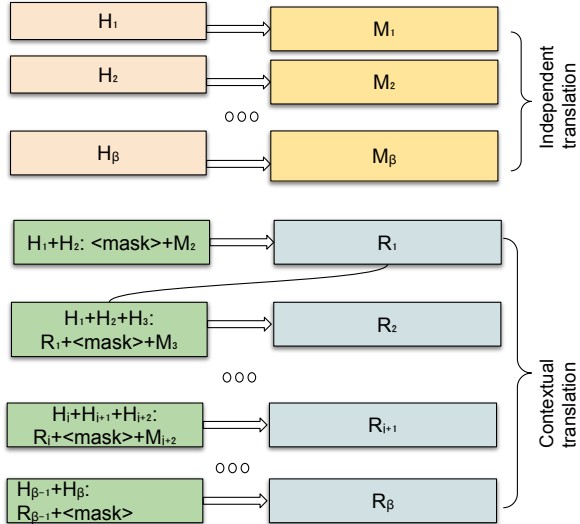

Figure 1: The diagram provides an overview of Decomposed Prompting for Machine Translation (DecoMT). The source text (H) is divided into several chunks ($H_1$, $H_2$,..,$H_i$,$H_{i+1}$,$H_{i+2}$,..,$H_\beta$). Each chunk is translated independently using few-shot prompting, yielding corresponding target chunks ($M_1$, $M_2$,..,$M_i$,$M_{i+1}$,$M_{i+2}$,..,$M_\beta$). The DecoMT process leverages the source chunks, their respective translations, and the previously predicted contextual translation to incrementally predict the contextually appropriate translation of the subsequent chunk.

technique dissects a complex task into simpler, more manageable subtasks, each of which is addressed through few-shot prompting of LLMs.

We aim to enhance translations by harnessing the inductive bias of monotonicity in related languages. We posit that by relieving LLMs from implicit reordering and focusing on sub-sentence structures, more accurate translations, particularly in longer sentences, can be achieved. This leads us to propose a decomposed prompting approach, termed Decomposed Prompting for Machine Translation (DecoMT) (Figure 1), which splits an input sentence into chunks, translates each independently, and incrementally generates context-aware translations.

While much of the existing research on prompting focuses on decoder-only LLMs, recent studies (Patel et al., 2023) show the potential of encoder-decoder models like mT5 (Xue et al., 2021) for such tasks. Our DecoMT approach builds upon this premise, utilizing the mT5 encoder-decoder LLM.

The following are our contributions:

- We introduce Decomposed Prompting for MT

(DecoMT), a novel approach that simplifies the translation task by dividing it into the incremental translation of word chunks.

- We perform extensive evaluations on closely related languages from diverse language families, including pairs such as Hindi ⇆ Marathi, Hindi ⇆ Malayalam, Hindi ⇆ Telugu, Hindi ⇆ Gujarati, Indonesian ⇆ Malay, Russian ⇆ Ukrainian, and Spanish ⇆ Portuguese.

- We compare DecoMT against several robust baselines, including few-shot prompting of LLMs (Lin et al., 2022; Workshop et al., 2023), as well as sequential autoregressive prompting of bidirectional LLMs (Patel et al., 2023). We demonstrate that DecoMT delivers robust results when compared to these baselines, particularly outperforming them in scenarios involving low-resource languages.

We release code and model outputs on github [1].

## 2 Related Work

**Few-shot Prompting for MT** Few-shot prompting for MT leverages an autoregressive LLM, which is prompted with a small number of sentence pairs alongside their translations. The LLM then predicts the translation when provided with a test sentence. Examples of such LLMs include XGLM (Lin et al., 2022) and BLOOM (Workshop et al., 2023). We interchangeably refer to this approach as Standard Prompting.

Garcia et al. (2023) have shown the effectiveness of few-shot prompting in machine translation. Yet, their method necessitates training a decoder-only LLM from scratch. In comparison, we use an off-the-shelf LLM, mT5, for DecoMT. A series of recent research delves into example selection for prompt construction (Vilar et al., 2023; Zhang et al., 2023; Kumar et al., 2023; Agrawal et al., 2023). In our method, we rely on a fixed set of examples for prompting. Jiao et al. (2023) analyzed machine translation using ChatGPT and found that ChatGPT's performance aligns closely with commercial translation systems when utilizing GPT-4. In the interest of reproducibility, our emphasis lies on publicly accessible LLMs like BLOOM and mT5.

---

[1] https://github.com/ratishsp/DecoMT

**Sequential Autoregressive Prompting** Patel et al. (2023) introduced an approach for prompting bidirectional LLMs, such as mT5 (Xue et al., 2021). Their Sequential Autoregressive Prompting (SAP) method generates a token autoregressively, appends it back to the input, and predicts the subsequent token. They demonstrated that SAP outperforms traditional few-shot prompting for LLMs. Our method also leverages bidirectional LLMs. However, while they primarily exploit the autoregressive nature of these models, we further harness the bidirectional capability of LLMs to generate context-aware translations.

**Decomposed Prompting** Khot et al. (2023) proposed decomposed prompting, an approach that breaks down complex tasks into simpler ones, each tackled using few-shot prompting of LLMs. We apply this prompting strategy to the task of machine translation between related languages.

**Incremental Generation** In the field of data-to-text generation, Puduppully et al. (2022) presented a strategy for document generation that decomposes the process into generating a sequence of paragraphs, interleaved with predicting a plan for each paragraph. Our DecoMT method can be viewed as an extension of this approach for the task of translating monotonically aligned sentences, where the plan is implicitly specified through the monotonic chunk alignment.

Press and Smith (2018) proposed an eager translation approach, in which the model begins translating without having to wait until the entire sentence has been processed. Our DecoMT method shares this characteristic, as it similarly doesn't require the whole sentence to be available before initiating translation. However, unlike their method, DecoMT's translation units extend beyond a single token. Moreover, DecoMT incorporates a contextual translation phase where the translation of an independent chunk is further refined through infilling.

**Machine Translation for Low Resource Languages** There have been studies on machine translation models for low-resource languages (Haddow et al., 2022; Team et al., 2022; Ramesh et al., 2022; AI4Bharat et al., 2023; Dabre et al., 2022). While most of these focus on translations between English and other languages, Fan et al. (2021) is notable for its emphasis on improving translations among non-English languages. Our

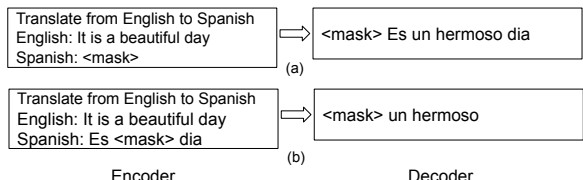

Figure 2: Depiction of two bidirectional encoder-decoder LLM prompting strategies for translation tasks. The upper part (a) uses an autoregressive translation, while part (b) employs the LLM for masked token infilling using surrounding context.

research aligns with this direction, concentrating on translations between related languages, many of which are characterized as low-resource.

## 3 DecoMT

In this section, we present the DecoMT Approach, our technique for decomposed prompting in Machine Translation. Our method involves a two-stage translation process for word chunks: firstly, an independent translation stage where each chunk is translated in isolation; and secondly, a contextual translation stage where translation occurs while considering the surrounding context.

### 3.1 Employed Pretrained Model

In implementing DecoMT, we use the mT5 model (Xue et al., 2021), specifically the XL variant with 3.7 billion parameters. mT5 is an encoder-decoder model that is trained with a span-corruption objective. During the training process of mT5, random spans within the input text are replaced with placeholders such as $\langle$mask_0$\rangle$, $\langle$mask_1$\rangle$, and so forth. In the output text, these correspond to mask tokens followed by the respective spans that were substituted in the input. Just like in the case of T5 (Raffel et al., 2020), the spans being replaced during training are of lengths varying from 2 to 5 tokens.

One approach to machine translation with mT5 follows the Standard Prompting method, as depicted in Figure 2 (a) (Workshop et al., 2023; Lin et al., 2022). In this setup, the mT5 encoder receives an input sequence: source language label, source sentence, target language label, followed by a $\langle$mask$\rangle$ token. The decoder then generates the translation. In our independent translation framework, we employ this technique to produce $M_i$ from $H_i$, as depicted in Figure 1.

Another technique to utilize mT5 for translation is by leveraging its bidirectional infilling capability,

as exhibited in Figure 2 (b). The prompt includes the source language label, source sentence, target language label and a partially masked translation. The mT5 decoder then generates the masked tokens. This specific approach is used in generating our contextual translations $R_i$ as shown in Figure 1.

Depending on where the $\langle mask \rangle$ placeholder is inserted, the model will perform either text completion or infilling. It's important to note that a single mask can yield more than one token.

## 3.2 Creating Aligned Monotonic Translations through Human Annotation

We select the first five examples from the dev set of the FLORES dataset (Goyal et al., 2022). Each example consists of a pair of corresponding sentences in two different languages. Annotators are tasked to align these sentences in a monotonic manner, maintaining the same sequence of information. Importantly, annotators have the liberty to modify the sentences as required to achieve this.

## 3.3 Translation Model

Let $x$ represent the input sentence and $\beta$ denote the number of chunks in $x$. We define $\hat{y}$ as the preliminary translation of $x$, obtained by concatenating independently translated chunks. Furthermore, $y$ represents the final translation, which is assembled from contextually translated chunks. For the purpose of simplification in our formulation, we omit the prompt template and focus on the translation of test examples.

In the case of independent translation, we make the assumption that each $\hat{y}_i$ is only dependent on its corresponding $x_i$, where $i$ indicates the index of the chunk within a sentence. This is captured by the equation:

$$p(\hat{y}|x) = \prod_{i=1}^{\beta} p(\hat{y}_i|x_i) \tag{1}$$

In the case of contextual translation, we parameterise $y$ as dependent on $x$ and $\hat{y}$, represented as:

$$p(y|x,\hat{y}) = p(y_1 y_2 \ldots y_\beta | x_1 x_2 \ldots x_\beta, \hat{y}_1 \hat{y}_2 \ldots \hat{y}_\beta) \tag{2}$$

We make a conditional independence assumption that, at any position $i$, $y_i$ is dependent on $x_{i-1}$, $x_i$, $x_{i+1}$, the previous contextual translation $y_{i-1}$, and the next independent translation $\hat{y}_{i+1}$. This assumption allows us to rewrite the joint probability

as a product of conditional probabilities:

$$
\begin{aligned}
p(y|x,\hat{y}) =\,& p(y_1|x_1 x_2 \hat{y}_2) \\
& * \prod_{i=2}^{\beta-1} p(y_i | x_{i-1} x_i x_{i+1} y_{i-1} \hat{y}_{i+1}) \\
& * p(y_\beta | x_{\beta-1} x_\beta y_{\beta-1})
\end{aligned}
$$

## 3.4 Prompt Construction

Our methodology employs few-shot prompting, a technique that allows an LLM to make predictions based on a limited number of examples. This section will elucidate the process of constructing prompts for independent and contextual translation. We utilize five examples for few-shot prompting.

**Word count in Each Chunk** Let us consider the token count within each word chunk in both prompt templates and test examples. For the prompt templates, $k$ and $j$ denote the number of tokens in a word chunk for independent and contextual translation, respectively. Conversely, in a test example, $m$ signifies the token count within a word chunk for independent translation.

We typically set $k$ and $j$ to 5 and 10, respectively. Nevertheless, the morphological richness of languages varies as a single token in one language might equate to several tokens in another. Hence, during the construction of prompt templates, we programmatically align each chunk fully with its translated equivalent, causing potential deviations from the standard values of 5 and 10 for $k$ and $j$.

Lastly, we treat $m$ as a hyperparameter, which is tuned using the FLORES development set.

**Independent Translation** Each translation example for independent translation (Figure 3) commences with "Translate from [Source language] to [Target language]:", followed by a line break, then "[Source language]:" and the first chunk of the source language sentence. Subsequently, we present "[Target language]:" and the corresponding translated chunk on a new line. This sequence is replicated for all the chunks in a sentence.

Upon completing a sentence, we use a newline separator and proceed to the next example. This procedure is repeated for all five examples in the prompt template.

In the case of the test example, the prompt begins with "Translate from [Source language] to [Target language]:", followed by a line break and "[Source language]:" with a chunk from the source language. The subsequent line is "[Target language]: $\langle mask \rangle$".

Translate from Hindi to Malayalam:
Hindi: सोमवार को, स्टैनफ़ोर्ड यूनिवर्सिटी स्कूल
Malayalam: തിങ്കളാഴ്ച്ച, സ്റ്റാൻഫോർഡ് യൂണിവേഴ്സിറ്റി സ്കൂൾ
(On Monday, Stanford University School)
Hindi:ऑफ़ मेडिसिन के वैज्ञानिकों ने
Malayalam:ഓഫ് മെഡിസിനിലെ ശാസ്ത്രജ്ഞന്മാർ
(of medicine scientists)
Hindi: कोशिकाओं को उनके प्रकार के
Malayalam: കോശങ്ങളെ അവയുടെ ഇനം
(cells into their types)
Hindi:आधार पर छाँट सकने वाला
Malayalam:അനുസരിച്ച് തരംതിരിക്കാൻ കഴിയുന്ന
(sort based on)
Hindi: एक नए डायग्नोस्टिक उपकरण के
Malayalam: ഒരു പുതിയ രോഗനിർണയ ഉപകരണം
(a new diagnostic tool)
Hindi:आविष्कार की घोषणा की.
Malayalam:കണ്ടുപിടിച്ചതായി പ്രഖ്യാപിച്ചു.
(announced the invention )

... 3 more examples here

Translate from Hindi to Malayalam:
Hindi: घटनास्थल की ओर जाते समय
Malayalam: സംഭവ സ്ഥലത്തേക്ക് പോകുന്ന സമയത്ത്
(on the way to the scene)
Hindi:एक एयरपोर्ट अग्निशामक वाहन लुढ़क गई ऐसा
Malayalam: ഒരു എയർപോർട്ട് ഫയർ വാഹനം കീഴ്മേൽ മറിഞ്ഞ–തായി
(an airport fire engine rolled over)
Hindi: स्थानीय मीडिया ने
Malayalam: പ്രാദേശിക മാധ്യമങ്ങൾ
(local media)
Hindi: बताया है.
Malayalam: റിപ്പോർട്ട്ചെയ്യുന.
(has told)

Translate from Hindi to Malayalam:
Hindi: कातलान की राजधानी (Catalan's capital)
Malayalam: <mask>

Figure 3: Prompt Template for Independent Translation with a Test Example: The template includes five sentences in the source (Hindi) and target (Malayalam) languages divided into word chunks. The model receives a test example source chunk and a target language prompt with a ⟨mask⟩ placeholder, aiming to predict the corresponding target chunk. English text in brackets is for clarification, not in the actual prompt.

Translate from Hindi to Malayalam:
Hindi: सोमवार को, स्टैनफ़ोर्ड यूनिवर्सिटी स्कूल ऑफ़ मेडिसिन के वैज्ञानिकों ने
Malayalam: തിങ്കളാഴ്ച്ച, സ്റ്റാൻഫോർഡ് യൂണിവേഴ്സിറ്റി സ്കൂൾ ഓഫ് മെഡിസിനിലെ ശാസ്ത്രജ്ഞന്മാർ
(On Monday, scientists at the Stanford University School of Medicine)
Hindi: कोशिकाओं को उनके प्रकार के आधार पर छाँट सकने वाला
Malayalam: കോശങ്ങളെ അവയുടെ ഇനം അനുസരിച്ച് തരംതിരി–ക്കാൻ കഴിയുന്ന
(capable of sorting cells according to their types)
Hindi: एक नए डायग्नोस्टिक उपकरण के आविष्कार की घोषणा की.
Malayalam: ഒരു പുതിയ രോഗനിർണയ ഉപകരണം കണ്ടുപിടിച്ച–തായി പ്രഖ്യാപിച്ചു.
(announced the invention of a new diagnostic tool)

... 3 more examples here

Translate from Hindi to Malayalam:
Hindi: घटनास्थल की ओर जाते समय एक एयरपोर्ट अग्निशामक वाहन लुढ़क गई ऐसा
Malayalam: സംഭവ സ്ഥലത്തേക്ക് പോകുന്ന സമയത്ത്ഒരു എയർ–പോർട്ട്ഫയർ വാഹനം കീഴ്മേൽ മറിഞ്ഞതായി
(an airport fire enginer rolled over on its way to the scene)
Hindi: स्थानीय मीडिया ने बताया है.
Malayalam: പ്രാദേശിക മാധ്യമങ്ങൾ റിപ്പോർട്ട്ചെയ്യുന.
(local media has told)

Translate from Hindi to Malayalam:
Hindi: कातलान की राजधानी (बार्सीलोना) में जाने के बाद से, विडाल ने क्लब के
(Since moving to the Catalan capital (Barcelona), Vidal has for the club) Malayalam: കാറ്റാലാണയുടെ തലസ്ഥാനമായ <mask> മുതല, വിദാൽ ക്ലബ്ബിന്
(Catalan's capital <mask> since Vidal for club)

Figure 4: Prompt Template for Contextual Translation with a Test Example: Similar to Figure 3, but with longer word chunks (approx. 10 tokens). The test prompt pairs a source language label with three concatenated word chunks. Following the target language label is the previous contextual translation, a ⟨mask⟩ placeholder, and the third chunk's independent translation. The model's goal is to complete the masked chunk. English bracketed text is explanatory and not a part of the prompt. The aligned chunks are colored identically.

The model's objective at this point is to predict the translation for the source language chunk.

**Contextual Translation** The prompt template for contextual translation (Figure 4) mirrors that of independent translation, with one key difference: the examples in prompt template are around twice as long as that of the lengths of examples in independent translation template prompt. In the test example for contextual translation, the prompt starts with "Translate from [Source language] to [Target language]:", followed by "[Source language]:" and a concatenation of three chunks from the source language.

The next line reads "[Target language]: [previous contextual translation] ⟨mask⟩ [next independent translation]". Here, the model's task is to infill the translation for the second source language chunk.

Appendix A contains an example of independent and contextual translation prompt templates for translation between Indonesian and Malay.

## 3.5 Inference

Figure 1 provides an overview of our DecoMT approach. We omit the prompt template from the block diagram for simplicity. We segment the input sentence into multiple chunks, denoted as $H_1$, $H_2$, ..., $H_i$, $H_{i+1}$, $H_{i+2}$, ..., $H_\beta$, each comprising $m$ tokens. We then independently translate each chunk into corresponding translations, labelled as $M_1$, $M_2$, ..., $M_i$, $M_{i+1}$, $M_{i+2}$, ..., $M_\beta$.

The key innovation in our approach lies in the contextual translation, which is performed incrementally for each chunk. Initially, we concatenate the first two chunks, $H_1$ and $H_2$, with the place-

holder ⟨mask⟩ and the translation of the second chunk $M_2$. This forms the input to predict the first contextual translation, $R_1$.

Subsequently, we concatenate the first three chunks, $H_1$, $H_2$, and $H_3$, with the contextual translation obtained from the previous step, $R_1$, alongside the placeholder ⟨mask⟩ and the translation of the third chunk, $M_3$. This is used to predict the next contextual translation, $R_2$.

This process is continued iteratively. At an intermediate step, the chunks $H_i$, $H_{i+1}$, and $H_{i+2}$, along with the previously computed contextual translation $R_i$, the placeholder ⟨mask⟩, and the translation of the chunk $M_{i+2}$, are used to predict the next contextual translation, $R_{i+1}$.

Finally, for the last chunk, the input is the concatenation of the penultimate and final chunks, $H_{\beta-1}$ and $H_\beta$, the last computed contextual translation $R_{\beta-1}$, and the placeholder ⟨mask⟩. The model then predicts the final contextual translation, $R_\beta$.

Appendix B contains a worked out example for translation from Hindi to Malayalam.

## 4 Experimental Setup

We conduct a comparative study of our DecoMT approach, which is based on mT5 (Xue et al., 2021) with 3.7B parameters, against various established approaches. These include the Standard Prompting technique applied to 7.1B parameters variant of BLOOM (Workshop et al., 2023), and 7.5B parameters variant of XGLM (Lin et al., 2022). We also compare our method with the Standard Prompting technique applied to the mT5 model. In this case, as mT5 generates only a few tokens at a time, we append the generated text back to the input to prompt further text generation. Furthermore, we compare our approach with SAP (Patel et al., 2023), a technique that also utilizes mT5 with 3.7B parameters.

### 4.1 Evaluation Metrics

Our approach's performance is assessed using spBLEU (Goyal et al., 2022), a variant of BLEU(Papineni et al., 2002), and chrF++ (Popović, 2017) metrics. The BLEU metric measures word n-gram matches, encompassing unigram, bigram, trigram, and four-grams. However, due to the morphological richness of the languages we are working with, BLEU scores can often be underestimated. To counteract this, we employ spBLEU as suggested by NLLB (Goyal et al., 2022; Team et al.,

2022), which utilizes a subword-based tokenizer.

Conversely, chrF++ evaluates character n-gram matches for n values ranging from 1 to 4, in addition to word n-gram matches that include unigram and bigram. Given its demonstrated higher correlation with human annotator scores for low-resource languages (Popović, 2017), chrF++ serves as a valuable metric for our study. We use the SacreBLEU library (Post, 2018) to compute these metrics. We provide signatures for both BLEU [2] and chrF++ [3].

For hyperparameter tuning, we utilize the FLORES development set. We evaluate chunk sizes for $m$ from the set {3,4,5}.

### 4.2 Evaluation

We conducted evaluations on multiple languages using the Flores devtest set, focusing specifically on translations between closely related languages: Hindi (hin) ↔ Marathi (mar), hin ↔ Malayalam (mal), hin ↔ Gujarati (guj), hin ↔ Telugu (tel), Indonesian (ind) ↔ Malay (zsm), Ukrainian (ukr) ↔ Russian (rus), and Portuguese (por) ↔ Spanish (spa). The latter pair represents a high-resource language setup for comparison.

## 5 Results

### 5.1 Automatic Evaluation

The results of our evaluations are summarized in Table 1. We conducted statistical significance testing via paired bootstrap sampling (Koehn, 2004) ($p < 0.05$). Regarding performance, XGLM (Lin et al., 2022) when used with Standard Prompting, demonstrated low spBLEU and chrF++ scores for low-resource language pairs such as hin↔mal, hin↔mar, hin↔guj, and ind↔zsm. It performed somewhat better with the ukr→rus pair, likely due to the greater availability of resources for Russian compared to Ukrainian.

BLOOM (Workshop et al., 2023), outperformed XGLM across all directions and language pairs except tel→hin. However, BLOOM does not currently support languages such as zsm, rus, and ukr.

When implemented with Standard Prompting, mT5 outperformed XGLM for most low-resource language pairs and even outperformed BLOOM on hin→mal, hin→guj, and hin→tel pairs, underscoring its effectiveness as a robust baseline.

---

[2] BLEU Signature: nrefs:1|case:mixed|eff:no| tok:flores200|smooth:exp|version:2.3.1

[3] chrF++ Signature: nrefs:1|case:mixed|eff:yes|nc:6| nw:2|space:no|version:2.3.1

| | spBLEU | | | | | chrF++ | | | | |
| | SP | | | SAP | DecoMT | SP | | | SAP | DecoMT |
| | BLOOM | XGLM | mT5 | mT5 | mT5 | BLOOM | XGLM | mT5 | mT5 | mT5 |
|---|---|---|---|---|---|---|---|---|---|---|
| hin→mal | 3.0 | 0.0 | 10.7 | 17.6 | **18.7** | 15.7 | 0.1 | 23.2 | 34.3 | **37.0** |
| mal→hin | 10.6 | 0.0 | 8.9 | 14.9 | **16.3** | 29.3 | 0.0 | 24.8 | 34.2 | **36.8** |
| hin→mar | 11.7 | 0.0 | 7.2 | 12.5 | **13.9** | 30.8 | 2.8 | 22.4 | 32.1 | **35.6** |
| mar→hin | 19.7[†] | 0.0 | 13.5 | 19.5 | **21.0** | 39.9 | 4.9 | 31.3 | 39.6 | **41.9** |
| hin→guj | 6.8 | 0.0 | 15.3 | 21.4 | **22.0** | 26.2 | 0.1 | 30.9 | 39.2 | **41.1** |
| guj→hin | 20.8 | 0.0 | 16.2 | 22.5 | **23.2** | 40.6 | 3.1 | 34.0 | 42.2 | **43.7** |
| hin→tel | 3.5 | 0.3 | 9.2 | 19.3[†] | **19.5** | 19.9 | 1.6 | 24.0 | 37.2 | **38.5** |
| tel→hin | 9.2 | 12.9 | 9.6 | 16.6 | **17.8** | 28.7 | 30.6 | 26.2 | 35.9 | **38.6** |
| zsm→ind | – | 0.0 | 18.1 | 28.7 | **29.6** | – | 7.4 | 40.8 | 53.9 | **55.9** |
| ind→zsm | – | 0.0 | 14.9 | 26.9 | **28.2** | – | 3.4 | 37.2 | 53.1 | **54.5** |
| rus→ukr | – | 5.7 | 19.2 | 30.1 | **31.0** | – | 24.3 | 36.4 | 48.0 | **49.9** |
| ukr→rus | – | 23.0 | 17.8 | 32.3 | **34.4** | – | 40.7 | 34.6 | 49.1 | **51.5** |
| spa→por | **29.1** | 28.3 | 13.6 | 27.6 | 26.5 | **51.5** | 49.4 | 32.0 | 48.4 | 50.0 |
| por→spa | **28.2** | 26.0 | 13.4 | 24.8 | 26.3 | **50.1** | 48.2 | 33.1 | 46.4 | 48.9 |

Table 1: The table presents spBLEU and chrF++ scores for standard prompting (SP) with BLOOM and XGLM, SAP with mT5, and our proposed DecoMT approach with mT5 across several language pairs, all tested on the FLORES devtest set. The highest performing results are highlighted in bold, and the second best scores are underlined for clarity. All comparisons with DecoMT demonstrate statistical significance ($p < 0.05$) (except results marked with [†]) as per paired bootstrap sampling (Koehn, 2004).

SAP proved to be a strong approach, echoing the findings of Patel et al. (2023). It outperformed Standard Prompting with BLOOM, XGLM and mT5 on the hin↔mal, hin↔mar, hin↔guj, hin↔tel, ind↔zsm, and rus↔ukr language pairs. Nevertheless, BLOOM outperformed SAP for the high-resource spa↔por pair.

Lastly, DecoMT surpassed all other approaches on the low-resource language pairs hin↔mal, hin↔mar, hin↔guj, hin↔tel, ind↔zsm, and rus↔ukr. While it also achieved impressive results with the high-resource spa↔por pair, it fell short of BLOOM's performance in this particular scenario. It's worth noting that DecoMT demonstrated an average improvement of 13.8 points in the chrF++ score over Standard Prompting with mT5, which presents a more direct comparison for DecoMT due to the same base model and their similar prompting and inference strategies.

## 5.2 Human Evaluation

To further analyze the quality of the outputs and validate the enhancements indicated by the automatic evaluation scores, we carry out a human evaluation study. This involves a comparative examination of our DecoMT approach, SAP, and Standard Prompting with mT5 and BLOOM.

We engaged annotators who possessed comprehension skills in the source language and demonstrated fluency in the target language. These annotators were remunerated in alignment with local hourly wage standards. The language pairs hin↔mar, hin↔guj, zsm→ind, and por→spa were selected for evaluation, contingent upon the availability of annotators well-suited for each pair. It should be noted that only a single annotator was assigned to each language pair. We sampled 50 sentences for each approach for a total of 200.

Our human evaluation strategy employs the Cross-Lingual Semantic Textual Similarity (XSTS) methodology (Licht et al., 2022) adopted by NLLB (Team et al., 2022) and IndicTrans2 (AI4Bharat et al., 2023). Within this approach, annotators are presented with the source sentence alongside translations produced by various approaches, omitting any human-annotated references. As XSTS emphasizes translation adequacy over fluency, it is well-suited to our focus on translation between related, typically low-resource languages, where adequacy takes precedence.

The XSTS metric is composed of a scale ranging from 1 to 5, where a score of 1 signifies completely dissimilar sentence pairs and a score of 5 represents semantically identical sentences. Appendix D contains details of the score values.

As shown in Table 2, DecoMT significantly out-

| | SP | | SAP | DecoMT |
|---|---|---|---|---|
| | BLOOM | mT5 | mT5 | mT5 |
| hin→mar | 2.4* | 1.9* | 2.3* | 3.0 |
| mar→hin | 3.4 | 2.3* | 3.4 | 3.6 |
| hin→guj | 2.0* | 2.1* | 3.4 | 3.2 |
| guj→hin | 3.0* | 2.1* | 3.3 | 3.6 |
| ind→zsm | 1.0* | 3.4* | 4.8 | 4.9 |
| por→spa | 4.7 | 2.5* | 4.1 | 4.5 |

Table 2: Human evaluation scores for standard prompting (SP) with BLOOM and XGLM, SAP with mT5, and our proposed DecoMT approach with mT5. Results marked with * indicate a statistically significant difference ($p < 0.05$) from DecoMT using ANOVA with post-hoc Tukey HSD test.

performs Standard Prompting with mT5 across all language pairs. DecoMT is significantly better than BLOOM for hin→mar, hin↔guj and ind→zsm but comparable with BLOOM on mar→hin and por→spa. DecoMT is significantly better than SAP for hin→mar, while demonstrating comparable performance for the remaining language pairs.

# 6 Discussion

**Scores of Translation across different Sentence Lengths** The DecoMT strategy involves translating source sentences in consecutive chunks, a method we hypothesize will lead to enhanced translation adequacy. To explore this, we group source sentences into length-based buckets, each with a width equivalent to the standard deviation of the source sentence lengths. If a bucket contains fewer than 20 instances, we merge it with its neighbour.

Figure 5 depicts the relationship between source sentence length and chrF++ scores for the hin→mal and zsm→ind language pairs. As hypothesized, as the length of the source sentence increases, the performance of DecoMT, as measured by chrF++, improves. For the zsm→ind language pair, the chrF++ scores of DecoMT and SAP are nearly identical for the first two buckets. However, as we move to the next three buckets with longer sentences, we observe a steady increase in DecoMT's chrF++ scores. This is in contrast with the declining scores of SAP, highlighting DecoMT's superiority in translating longer sentences.

**Improvement by Adding the Contextual Translation Compared to the Independent Translation** We compared the single-stage independent translation to the two-stage DecoMT. The experiments show that the inclusion of contextual transla-

tion in the second stage of DecoMT significantly improves performance. We report the improvement in chrF++ scores in Table 3. The improvement in spBLEU is presented in Appendix E.

| Lang Pair | chrF++ (Single Stage) | Δ chrF++ |
|---|---|---|
| hin->mal | 33.7 | +3.3 |
| mal->hin | 34.7 | +2.1 |
| hin->mar | 33.1 | +2.5 |
| mar->hin | 39.6 | +2.3 |
| hin->guj | 39.6 | +1.5 |
| guj->hin | 41.8 | +1.9 |
| hin->tel | 36.3 | +2.2 |
| tel->hin | 35.7 | +2.9 |
| zsm->ind | 53.8 | +2.1 |
| ind->zsm | 54.3 | +0.2 |
| rus->ukr | 48.5 | +1.4 |
| ukr->rus | 49.9 | +1.6 |
| spa->por | 48.7 | +1.3 |
| por->spa | 47.1 | +1.8 |

Table 3: Improvement in chrF++ scores gained by the DecoMT approach compared to the Single Stage.

**Off-target Translations** To quantify the off-target translation rate among various approach's outputs, we employed the Language Identification tool developed by the NLLB (Team et al., 2022). The off-target translation rate is represented as a percentage, with a lower percentage denoting superior performance, as shown in Table 4. We see that the DecoMT approach consistently outperforms other approaches with lower off-target translation rate across various translation tasks. We conduct further analysis in Appendix F.

**Extension to Autoregressive and other Encoder-Decoder LLMs** At present, we utilize mT5 for both independent and contextual translations. However, it's worth noting that any autoregressive LLM could potentially be used for independent translation. As for contextual translation, an autoregressive LLM could be prompted with a fill-in-the-blanks type of prompt - an avenue we intend to explore in future work. Additionally, the exploration of other encoder-decoder LLMs such as UL2 (Tay et al., 2023) or AlexaTM (Soltan et al., 2022) for contextual translations presents a promising research direction.

**Experiments with Zero-shot and One-shot Prompting** We undertook zero-shot translation experiments for select language pairs, specifically

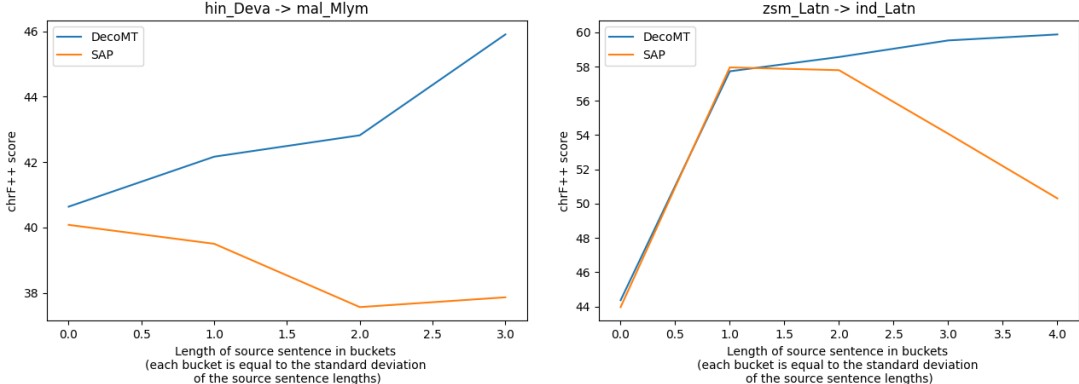

Figure 5: The plots show the relationship between source sentence length and chrF++ scores for hin→mal and zsm→ind pairs. Lengths are bucketed, each equal to the sentence lengths' standard deviation, with any bucket with less than 20 sentences merged with its neighbour. The data implies DecoMT's chrF++ scores outperform SAP's with increasing sentence length, indicating DecoMT's proficiency with longer sentences.

| | SP | | | SAP | DecoMT |
|---|---|---|---|---|---|
| | BLOOM | XGLM | mT5 | mT5 | mT5 |
| hin→mal | 23.6 | 100.0 | 14.4 | 0.4 | 0.0 |
| mal→hin | 8.4 | 0.0 | 4.4 | 1.4 | 0.2 |
| hin→mar | 21.2 | 96.3 | 35.2 | 10.0 | 0.8 |
| mar→hin | 1.3 | 20.0 | 2.6 | 1.1 | 0.2 |
| hin→guj | 10.2 | 99.7 | 3.8 | 0.2 | 0.0 |
| guj→hin | 3.3 | 0.0 | 1.9 | 0.4 | 0.2 |
| zsm→ind | – | 48.8 | 23.3 | 17.7 | 13.1 |
| ind→zsm | – | 94.2 | 59.7 | 47.3 | 30.1 |
| rus→ukr | – | 84.3 | 1.7 | 0.2 | 0.0 |
| ukr→rus | – | 0.6 | 0.5 | 0.1 | 0.0 |
| spa→por | 0.2 | 0.4 | 3.4 | 0.9 | 0.2 |
| por→spa | 0.0 | 0.5 | 0.6 | 0.3 | 0.1 |

Table 4: The percentage of sentences off-target for a translation direction. Lower is better.

hin<->guj, hin<->tel, and hin<->mal. We compared different approaches applied to mT5 including DecoMT, SAP and Standard Prompting. We found that all approaches yielded near-zero BLEU scores. In most instances, the models merely copied the input as the output. We hypothesize that this is because in a zero-shot setting the model may not understand that it has to perform translation to the target language.

We compared one-shot and five-shot settings for three language pairs (hin<->guj, hin<->tel and hin<->mal) using Standard Prompting (SP), SAP, and DecoMT with mT5. Our results in Appendix G indicate that:

- DecoMT maintains strong performance even in the one-shot setting.

- Both SAP and SP experience significant performance drops transitioning from five-shot to one-shot. For instance, the spBLEU score for hin->tel in SAP drops from 19.3 (five-shot) to just 1.3 (one-shot).

**Inference Times** As highlighted in Patel et al. (2023), to generate a sentence comprising $T$ words, SAP necessitates $T$ forward passes through the model. This approach stands in contrast to Standard Prompting, which only requires a single pass. In the case of DecoMT, the independent translation stage can be parallelized with relative ease. For the contextual translation stage, $T/m$ forward passes through the model are needed, where $m$ denotes the chunk size. As a result, the inference time for DecoMT is less than that of SAP. Appendix H contains more details of runtime analysis.

## 7 Conclusion

In this study, we introduced DecoMT, a novel approach using decomposed prompting for Machine Translation of related languages. DecoMT demonstrated superior performance over established few-shot prompting baselines in translating between low-resource related languages, as evidenced by our experiments with the FLORES dataset. Additionally, DecoMT showed robust performance even in high-resource scenarios.

## Limitations

Despite its advantages, DecoMT does possess certain limitations. Notably, the approach requires human annotation for constructing the five example-aligned prompts in the template. However, our

observations suggest that the annotators primarily need to modify existing translations, which is less laborious than generating translations from scratch, an activity that can be done in under 30 minutes. Conversely, other baseline approaches don't require such annotation and are able to directly utilize translation examples.

When considering the translation time, DecoMT, given its two-stage process encompassing independent and contextual translations, inherently requires a longer duration to generate outputs compared to traditional few-shot prompting methodologies.

Another limitation of DecoMT is its dependency on an LM with infixing capabilities during the contextual translation stage. In the absence of infixing capabilities, this can be simulated on other LLM with appropriate prompting, and we plan to explore that in future work.

## Ethics Statement

This study does not involve any new data collection. We solely utilize publicly accessible datasets for conducting the experiments reported herein. Furthermore, for the purpose of annotation of translation examples and the human evaluation of machine translation outputs, we employ annotators who are duly compensated for their time and expertise, ensuring fair practices in line with established standards.

## Acknowledgements

We would like to thank the reviewers for their feedback. This research was supported by funding from the Institute for Infocomm Research (I2R) under A*STAR ARES, Singapore. We extend our gratitude to Litton Kurisinkel, Aswanth Kumar, Siti Umairah, Ivan Kukanov, Swapnali Waghunde, and Fabian Ritter-Gutierrez for their work in annotating the few-shot prompts. Additionally, we'd like to thank Siti Umairah, Fabian Ritter-Gutierrez, Kunal Gandhi, and Faiz Masi for their contributions to the human evaluation experiments.

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

## A Examples of Prompts

The prompts used for independent and contextual translations by DecoMT for the language pair Malay→Indonesian are presented in Table 5 and Table 6, respectively. Meanwhile, Table 7 illustrates the prompts utilized for Standard Prompting and SAP.

Translate from Malay to Indonesian:
Malay: Saintis dari Stamford Universiti Sekolah
Indonesian: Ilmuwan dari Stanford University School of
Malay: Perubatan pada hari Isnin
Indonesian: Medicine pada hari Senin
Malay: mengumumkan penemuan alat diagnostik baharu
Indonesian: mengumumkan penemuan alat diagnostik baru
Malay: yang boleh menyusun sel berdasarkan
Indonesian: yang bisa mengurutkan sel berdasarkan
Malay: jenis: cip kecil dapat dicetak
Indonesian: tipe: cip kecil dapat dicetak
Malay: yang boleh dihasilkan menggunakan printer
Indonesian: yang bisa diproduksi menggunakan printer
Malay: inkjet standard dengan kos sekitar
Indonesian: inkjet standar dengan biaya sekitar
Malay: satu sen AS se cip.
Indonesian: satu sen AS per cip.

Translate from Malay to Indonesian:
Malay: Ketua penyelidik mengatakan bahawa diagnosis
Indonesian: Ketua peneliti mengatakan bahwa diagnosis
Malay: ini mungkin dapat menghasilkan pengesanan
Indonesian: ini mungkin dapat menghasilkan deteksi
Malay: awal kanser, tuberkulosis, HIV, dan
Indonesian: dini kanker, tuberkulosis, HIV, dan
Malay: malaria kepada pesakit-pesakit di negara
Indonesian: malaria kepada pasien-pasien di negara
Malay: berpendapatan rendah, di mana kadar
Indonesian: berpenghasilan rendah, di mana tingkat
Malay: kesembuhan dari penyakit-penyakit seperti kanser
Indonesian: kesembuhan dari penyakit-penyakit seperti kanker
Malay: payudara boleh mencapai setengah dari
Indonesian: payudara bisa mencapai setengah dari
Malay: negara-negara kaya.
Indonesian: negara-negara kaya.

Translate from Malay to Indonesian:
Malay: JAS 39C Gripen terhempas ke
Indonesian: JAS 39C Gripen jatuh ke
Malay: landasan sekitar jam 9:30
Indonesian: landasan pacu sekitar pukul 9:30
Malay: waktu tempatan (0230 UTC) dan
Indonesian: waktu setempat (0230 UTC) dan
Malay: meletup, mengakibatkan ditutup lapangan terbang
Indonesian: meledak, menyebabkan ditutupnya bandara
Malay: untuk penerbangan komersial.
Indonesian: untuk penerbangan komersial.

Translate from Malay to Indonesian:
Malay: Juruterbang tersebut dikenalpasti sebagai Ketua
Indonesian: Pilot tersebut diidentifikasi sebagai Pemimpin
Malay: Pasukan Dilokrit Pattavee.
Indonesian: Skuadron Dilokrit Pattavee.

Translate from Malay to Indonesian:
Malay: Media tempatan melaporkan sebuah kenderaan
Indonesian: Media lokal melaporkan sebuah kendaraan
Malay: pemadam api di lapangan terbang tergolek
Indonesian: pemadam api di bandara terguling
Malay: ketika dikendalikan.
Indonesian: saat sedang dioperasikan.

Translate from Malay to Indonesian:

Table 5: Prompt for Independent translation in DecoMT from Malay to Indonesian

Translate from Malay to Indonesian:
Malay: Saintis dari Stamford Universiti Sekolah Perubatan pada hari Isnin
Indonesian: Ilmuwan dari Stanford University School of Medicine pada hari Senin
Malay: mengumumkan penemuan alat diagnostik baharu yang boleh menyusun sel berdasarkan
Indonesian: mengumumkan penemuan alat diagnostik baru yang bisa mengurutkan sel berdasarkan
Malay: jenis: cip kecil dapat dicetak yang boleh dihasilkan menggunakan printer
Indonesian: tipe: cip kecil dapat dicetak yang bisa diproduksi menggunakan printer
Malay: inkjet standard dengan kos sekitar satu sen AS se cip.
Indonesian: inkjet standar dengan biaya sekitar satu sen AS per cip.

Translate from Malay to Indonesian:
Malay: Ketua penyelidik mengatakan bahawa diagnosis ini mungkin dapat menghasilkan pengesanan
Indonesian: Ketua peneliti mengatakan bahwa diagnosis ini mungkin dapat menghasilkan deteksi
Malay: awal kanser, tuberkulosis, HIV, dan malaria kepada pesakit-pesakit di negara
Indonesian: dini kanker, tuberkulosis, HIV, dan malaria kepada pasien-pasien di negara
Malay: berpendapatan rendah, di mana kadar kesembuhan dari penyakit-penyakit seperti kanser
Indonesian: berpenghasilan rendah, di mana tingkat kesembuhan dari penyakit-penyakit seperti kanker
Malay: payudara boleh mencapai setengah dari negara-negara kaya.
Indonesian: payudara bisa mencapai setengah dari negara-negara kaya.

Translate from Malay to Indonesian:
Malay: JAS 39C Gripen terhempas ke landasan sekitar jam 9:30
Indonesian: JAS 39C Gripen jatuh ke landasan pacu sekitar pukul 9:30
Malay: waktu tempatan (0230 UTC) dan meletup, mengakibatkan ditutup lapangan terbang
Indonesian: waktu setempat (0230 UTC) dan meledak, menyebabkan ditutupnya bandara
Malay: untuk penerbangan komersial.
Indonesian: untuk penerbangan komersial.

Translate from Malay to Indonesian:
Malay: Juruterbang tersebut dikenalpasti sebagai Ketua Pasukan Dilokrit Pattavee.
Indonesian: Pilot tersebut diidentifikasi sebagai Pemimpin Skuadron Dilokrit Pattavee.

Translate from Malay to Indonesian:
Malay: Media tempatan melaporkan sebuah kenderaan pemadam api di lapangan terbang tergolek
Indonesian: Media lokal melaporkan sebuah kendaraan pemadam api di bandara terguling
Malay: ketika dikendalikan.
Indonesian: saat sedang dioperasikan.

Translate from Malay to Indonesian:

Table 6: Prompt for Contextual translation in DecoMT from Malay to Indonesian

Translate from Malay to Indonesian:
Malay: Pada hari Isnin, Saintis daripada Sekolah Perubatan Universiti Stamford mengumumkan penemuan alat diagnostik baru yang boleh mengasingkan sel-sel mengikut jenis: cip kecil yang boleh dicetak yang boleh dihasilakn menggunakan pencetak standard inkjet untuk kira-kira satu sen A.S setiap satu.
Indonesian: Ilmuwan dari Stanford University School of Medicine pada hari Senin mengumumkan penemuan alat diagnostik baru yang bisa mengurutkan sel berdasarkan tipe: cip kecil dapat dicetak yang bisa diproduksi menggunakan printer inkjet standar dengan biaya sekitar satu sen AS per cip.

Translate from Malay to Indonesian:
Malay: Penyelidik utama mengatakan bahawa ia mungkin menghasilkan pengesanan awal kanser, tuberkulosis, HIV dan malaria kepada pesakit di negara-negara berpendapatan rendah, di mana kadar kemandirian untuk penyakit seperti kanser payu dara ialah separuh daripada di negara-negara yang lebih kaya.
Indonesian: Ketua peneliti mengatakan bahwa diagnosis ini mungkin dapat menghasilkan deteksi dini kanker, tuberkulosis, HIV, dan malaria kepada pasien-pasien di negara berpenghasilan rendah, di mana tingkat kesembuhan dari penyakit-penyakit seperti kanker payudara bisa mencapai setengah dari negara-negara kaya.

Translate from Malay to Indonesian:
Malay: JAS 39C Gripen telah terhempas ke atas landasan sekitar jam 9:30 pagi waktu tempatan (0230 UTC) dan meletup, mengakibatkan lapangan terbang ditutup bagi penerbangan komersial.
Indonesian: JAS 39C Gripen jatuh ke landasan pacu sekitar pukul 9.30 waktu setempat (0230 UTC) dan meledak, menyebabkan ditutupnya bandara untuk penerbangan komersial.

Translate from Malay to Indonesian:
Malay: Juruterbang telah dikenal pasti sebagai Ketua Pasukan Dilokrit Pattavee.
Indonesian: Pilot tersebut diidentifikasi sebagai Pemimpin Skuadron Dilokrit Pattavee.

Translate from Malay to Indonesian:
Malay: Media tempatan melaporkan kenderaan api lapangan terbang terguling ketika memberi maklum balas.
Indonesian: Media lokal melaporkan sebuah kendaraan pemadam api di bandara terguling saat sedang dioperasikan.

Translate from Malay to Indonesian:

Table 7: Prompt for Standard Prompting and SAP from Malay to Indonesian

# B   Example for DecoMT

Figure 6 presents a block diagram which explains DecoMT with the help of an example. The task at hand is translation from Hindi to Malayalam. The Hindi sentence is divided into four consecutive chunks: $H_1$, $H_2$, $H_3$, and $H_4$, each consisting of $m = 5$ tokens. Using few-shot prompting, these chunks are independently translated into Malayalam, resulting in $M_1$, $M_2$, $M_3$, and $M_4$. However,

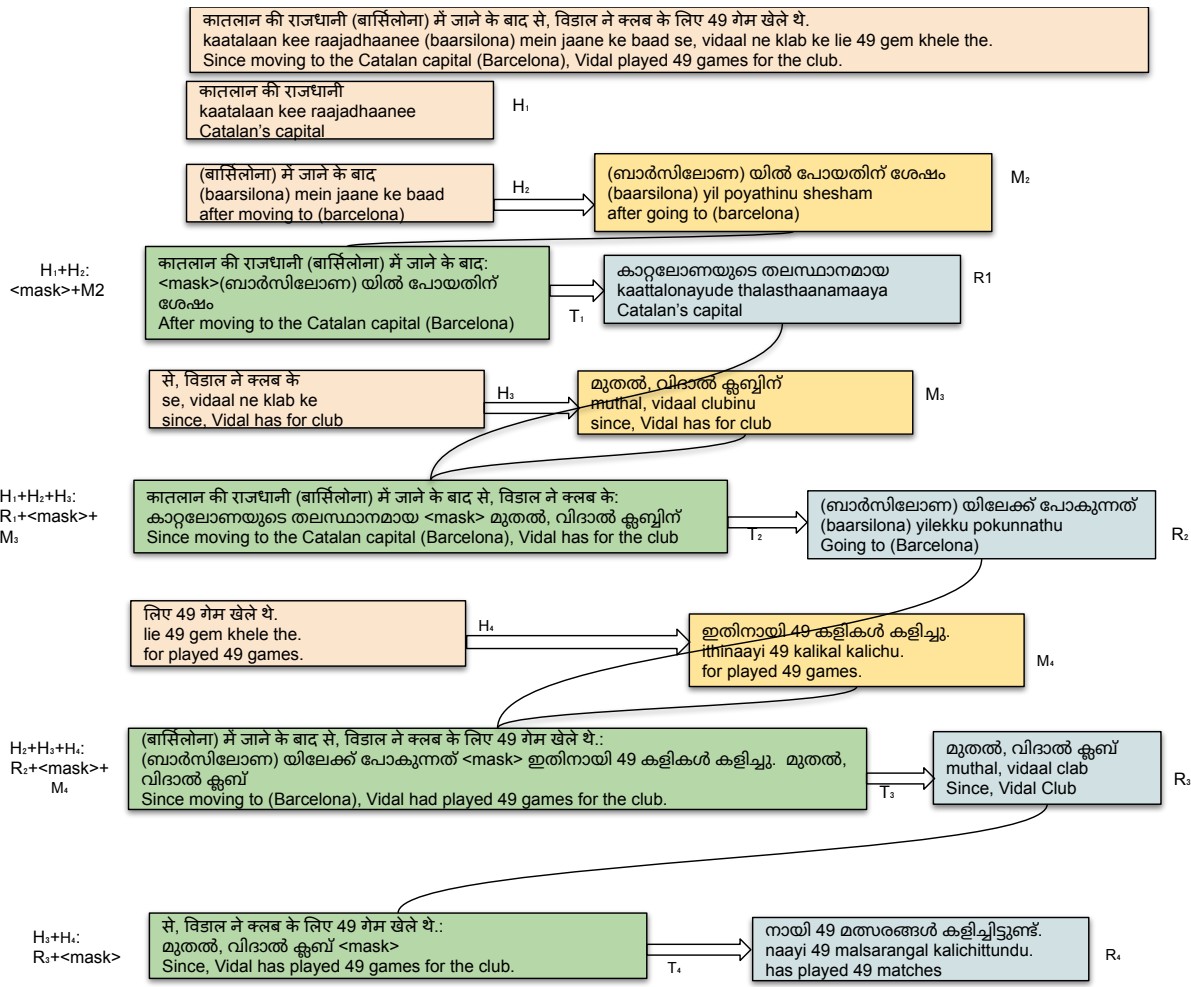

Figure 6: This diagram provides a step-by-step illustration of the DecoMT process. For the sake of simplifying our explanation, we have excluded the prompt template from the block diagram. The chunks of Hindi input, represented as $H_1$, $H_2$, $H_3$, and $H_4$, are initially translated into Malayalam independently using few-shot prompting, resulting in $M_1$, $M_2$, $M_3$, and $M_4$. Subsequently, infilling is used to derive contextual translations, denoted as $R_1$, $R_2$, $R_3$, and $R_4$. Each block of $H_i$, $M_i$, and $R_i$ presents three lines: the original text, its English transliteration, and its translation into English. The blocks marked $T_i$ illustrate the contextual translation tasks. The input block for $T_i$ includes a concatenation of input chunks, the previous contextual translation, a mask placeholder, and an independent translation, along with their English translation. The final translation into Malayalam, is produced by piecing together the contextual translations $R_1$, $R_2$, $R_3$, and $R_4$. It should be noted that the English translations and transliterations are included for the sake of clarity and are not an integral part of the DecoMT process.

we observe that these translated chunks can occasionally lack coherence.

For instance, consider the translation of the $H_4$ chunk. The chunk commences with लिए which can translate to 'reason' or 'for' (indicating possession) in English. The $M_4$ translation into Malayalam, ഇതിനായി adopts the former meaning, whereas the sentence context implies that the latter interpretation would be more suitable.

To rectify this, we introduce a process to generate contextually appropriate translations. We input a concatenation of $H_1$, $H_2$, and a mask placeholder, along with $M_2$, into the bidirectional mT5 model.

The model then infills the mask, producing a contextually appropriate translation of $M_1$, which we denote as $R_1$.

Next, we feed a concatenation of $H_1$, $H_2$, $H_3$, along with a concatenation of $R_1$, a mask placeholder, and $M_3$ into the mT5 model. The result is a contextually appropriate translation, $R_2$, of $M_2$.

This procedure is repeated for all the intermediate chunks. For the final chunk, we input a concatenation of $H_3$, $H_4$, $R_3$, and a mask placeholder. The mT5 model then predicts the contextually appropriate translation, $R_4$, of the $M_4$ translation. Given the context of $H_3$, $H_4$, and $R_3$, the contextual trans-

| Language pair | $m$ |
|---|---|
| hin→mal | 5 |
| mal→hin | 3 |
| hin→mar | 5 |
| mar→hin | 4 |
| hin→guj | 5 |
| guj→hin | 4 |
| hin→tel | 5 |
| tel→hin | 3 |
| zsm→ind | 4 |
| ind→zsm | 4 |
| rus→ukr | 4 |
| ukr→rus | 4 |
| por→spa | 4 |
| spa→por | 4 |

Table 8: Optimum value of m found through hyperparameter search in {3,4,5}.

lation correctly interprets the intended meaning.

## C Hyperparameter $m$

The optimum value of $m$ for different language pairs is presented in Table 8. We posit that the optimal value of $m$ is contingent on the relative morphological complexity of the source language. Take the example of hin↔mal. Since Hindi (hin) is less morphologically complex than Malayalam (mal), a larger number of tokens are required in a chunk for hin→mal than for mal→hin to produce satisfactory outputs in the independent translation stage.

In the case of zsm↔ind, both languages exhibit similar morphological complexity, resulting in an identical optimum value of $m$, which is 4. The same applies to the rus↔ukr and spa↔por pairs. For these three pairs, a value of $m$ smaller than 4 results in subpar independent translation quality. Conversely, a value exceeding 4 might lead to truncated translations.

## D Details of Human Annotation Guidelines

The XSTS metric provides ratings between 1 and 5, representing different levels of similarity between sentences.

- A score of 1 indicates that the sentences share little content or may be about different topics. If they share content, it is less than 50

- A score of 2 indicates that the sentences are about similar topics but are not equivalent, and

there may be differences in important information related to the primary subject/verb/object.

- A score of 3 indicates that the sentences are mostly similar, but there may be some minor omissions of unimportant information. There should not be any significant conflict in the information.

- A score of 4 indicates that the sentences are paraphrases of each other. There are no major differences or missing information, although there may be variations in expression such as tone, style, emphasis, or formality.

- A score of 5 indicates that the sentences are completely equivalent in meaning and usage, including expression aspects such as formality, tones, style, and emphasis.

For more details and examples, see Licht et al. (2022).

## E Improvement by Adding the Contextual Translation Compared to the Independent Translation

Table 9 showcases the improvements in spBLEU scores achieved by the DecoMT approach in comparison to the Single Stage method.

| Lang Pair | spBLEU (Single Stage) | Δ spBLEU |
|---|---|---|
| hin->mal | 15.9 | +2.8 |
| mal->hin | 13.3 | +3.0 |
| hin->mar | 12.1 | +1.8 |
| mar->hin | 17.0 | +4.0 |
| hin->guj | 20.2 | +1.8 |
| guj->hin | 21.0 | +2.2 |
| hin->tel | 16.7 | +2.8 |
| tel->hin | 11.3 | +6.5 |
| zsm->ind | 26.4 | +3.2 |
| ind->zsm | 27.7 | +0.5 |
| rus->ukr | 28.3 | +2.7 |
| ukr->rus | 32.1 | +2.3 |
| spa->por | 24.4 | +2.1 |
| por->spa | 23.7 | +2.6 |

Table 9: Improvement in spBLEU scores gained by the DecoMT approach compared to the Single Stage.

## F Off-target Translations

In Table 4, focusing on the relatively high off-target translation rate for ind↔zsm, particularly for ind→zsm, we analyzed 50 mislabeled DecoMT

| Language Pair | spBLEU | | | chrF++ | | |
|---|---|---|---|---|---|---|
| | SP mT5 | SAP mT5 | DecoMT mT5 | SP mT5 | SAP mT5 | DecoMT mT5 |
| hin->guj one-shot | 10.0 | 16.1 | 22.2 | 20.5 | 29.6 | 41.1 |
| hin->guj five-shots | 15.3 | 21.4 | 22.0 | 30.9 | 39.2 | 41.1 |
| guj->hin one-shot | 17.1 | 22.9 | 23.0 | 34.7 | 42.7 | 43.4 |
| guj->hin five-shots | 16.2 | 22.5 | 23.2 | 34.0 | 42.2 | 43.7 |
| hin->tel one-shot | 0.4 | 1.3 | 18.9 | 1.5 | 2.8 | 38.2 |
| hin->tel five-shots | 9.2 | 19.3 | 19.5 | 24.0 | 37.2 | 38.5 |
| tel->hin one-shot | 5.3 | 8.7 | 17.4 | 12.6 | 18.6 | 38.4 |
| tel->hin five-shots | 9.6 | 16.6 | 17.8 | 26.2 | 35.9 | 38.6 |
| hin->mal one-shot | 1.3 | 2.9 | 18.2 | 3.2 | 5.7 | 36.7 |
| hin->mal five-shots | 10.7 | 17.6 | 18.7 | 23.2 | 34.3 | 37.0 |
| mal->hin one-shot | 9.1 | 13.4 | 16.5 | 22.7 | 29.8 | 36.9 |
| mal->hin five-shots | 8.9 | 14.9 | 16.3 | 24.8 | 34.2 | 36.8 |

Table 10: Comparison of one-shot and five-shot translation results across three language pairs using SP, SAP, and DecoMT with mT5. Notably, DecoMT exhibits robust performance in one-shot settings, whereas SP and SAP show marked performance reductions, exemplified by the spBLEU drop for hin->tel in SAP from 19.3 (five-shot) to 1.3 (one-shot).

output sentences from ind→zsm. An annotator from our human evaluation study (Section 5.2) found that 64% of these sentences were in fact Malay, not Indonesian. This suggests potential shortcomings in automatic language identification for closely related languages such as ind and zsm.

## G Comparison between One-shot and Five-shot Prompting

As detailed in Table 10, our evaluations span three language pairs and compare the efficacy of Standard Prompting (SP), SAP, and DecoMT methodologies when evaluated on mT5. In comparison between one-shot and five-shot scenarios, we find that DecoMT consistently demonstrates strong performance in one-shot settings, in contrast to the pronounced performance dips observed for both SP and SAP.

## H Analysis of Runtime

To ensure a fair comparison, we profile the codes using cprofile [4] during the inference phase, executed on an A40 48GB GPU. cprofile examines the time taken by various API calls. In this case, our chosen task is translating from Marathi to Hindi using the initial batch of 5 examples from the FLO-RES test set, with the longest Marathi sample in the batch being 41 tokens long.

---
[4]https://docs.python.org/3/library/profile.html

• SAP Analysis: For the SAP system, due to the unpredictability of the expected target length, we do decoding at 1.5 times the maximum source length. This is based on our studies of lengths of examples from validation dataset. For example, for our given source batch, the reference Hindi translation encompasses 55 tokens for the Marathi sentence which is 41 tokens long. As the longest example is 41 tokens, we run inference for 41 * 1.5 = 61 steps. Table 11 contains a partial trace of performance profiling using cprofile. We see that for SAP, there are 61 calls to predict_output method. The method predict_output is responsible for running inference on the LLM. Each method takes 2.384 seconds. The inference of the batch takes 145.455 seconds.

| ncalls | cumtime | percall | filename:lineno(function) |
|---|---|---|---|
| 3/1 | 145.455 | 145.455 | {built-in method builtins.exec} |
| ... | ... | ... | ... |
| 61 | 145.428 | 2.384 | sap.py:163 (predict_output) |

Table 11: Performance Profiling Data for SAP

• DecoMT Analysis: For Marathi-Hindi translations, we use a chunk size of 4. We first consider the independent translation stage. Breaking down the sentence lengths of the batch in tokens: 16, 30, 24, 41, and 28, we get respective chunk counts of 4, 8, 6, 11, and 7—ag-

gregating to 36 chunks. Split into batches of 8, this leads to 5 API calls to predict_output. With the longest sentence in the batch having 41 tokens, the contextual translation stage demands 11 API calls to predict_output, cumulating to 16 calls. These 16 api calls in total amount to 96.868 seconds (Table 12). While predict_output in DecoMT tends to take longer than in SAP (owing to DecoMT predicting multiple tokens as opposed to SAP's single-token approach), the overall fewer API calls render DecoMT more efficient.

| ncalls | cumtime | percall | filename:lineno(function) |
|--------|---------|---------|---------------------------|
| 3/1 | 96.883 | 96.883 | `{built-in    method builtins.exec}` |
| ... | ... | ... | ... |
| 16 | 96.868 | 6.054 | `decomt.py:199 (predict_output)` |

Table 12: Performance Profiling Data for DecoMT