# OpenReview forum: "DecoMT: Decomposed Prompting for Machine Translation Between Related Languages using Large Language Models"
_EMNLP/2023/Conference — EMNLP 2023 Main_

### Official Review · Reviewer_C2ZS · 2023-08-02

**Soundness:** 3

**Excitement:**

3: Ambivalent: It has merits (e.g., it reports state-of-the-art results, the idea is nice), but there are key weaknesses (e.g., it describes incremental work), and it can significantly benefit from another round of revision. However, I won't object to accepting it if my co-reviewers champion it.

**Missing References:**

Since there are quite a lot papers working prompting LLMs for machine translation that appeared quickly this year, it would be better to mention some of them appeared three months before the submission, even though they may not be officially reviewed:

- The unreasonable effectiveness of few-shot learning for machine translation. Garcia et al., 2023
- Prompting PaLM for Translation: Assessing Strategies and Performance. Vilar et al., 2022
- Prompting Large Language Model for Machine Translation: A Case Study. Zhang et al., 2023
- Is ChatGPT A Good Translator? A Preliminary Study. Jiao et al., 2023

**Paper Topic And Main Contributions:**

This paper propose a two-pass decoding method to use a mT5 model for direct translation between related languages.

During the first decoding round, the authors split a source sentence into several chunks and performs independent translation of each chunk. (H_i -> M_i) In the second decoding round, each source chunk is concatenated with the neighbor chunks (left and right) as context (H_i-1 H_i H_i+1), and the corresponding translated context segment from round 1 is reused as target context (M_i-1 <mask> M_i+1). The model then performs a text in-filling task to output the translation correponds to the <mask> in the given context (H_i-1 H_i H_i+1: R_i-1 <mask> M_i+1 -> R_i). Finally the generated R_i are concatenated as the final translation. The translation process is performed under a few-shot prompting manner, where the source and target language is indicated in the prompting template.

Experiments show that the proposed method deliver better translation results than previous prompting methods using mT5 on several related language pairs, especially when translation longer sentences. LLMs are known to have limited performance on low resource languages, especially translation between non-English languages. This paper gives a solution to perform direct translation between related non-English languages without passing through English using LLMs.

However, this paper could still benefit from some more experiments and analyses to better demonstrate the effectiveness of the proposed method. For example, this paper only compares few-shot prompting performance. As all compared methods, including the proposed method, seem applicable in zero-shot scenarios, it is better to see the effectiveness of using few-shot examples. Also, since the final translation only comes from the second decoding round, what is the performance of the first decoding round? Does the second round really improves the translation quality? Is it necessary?

**Questions For The Authors:**

- L449, the authors mentioned that each language pair only have one single annotator. Does this cause randomness in the human evaluation results?

**Reasons To Accept:**

- The proposed two-pass decoding method is interesting. It reuses the generated translation through independent chunk translation as pseudo target context to perform the second decoding round. It takes the advantage of bidirectional LM to perform both direct translation and text in-filling, which may not be possible for unidirectional LM if not specifically fine-tuned.
- The proposed method is effective. It delivers better translation performance than previous prompting methods based on both unidirectional LM and bidirectional LM over several language pairs.
- This proposed method may be an inspiration for future work to perform translation between non-English or low-resource languages using LLMs.

**Reasons To Reject:**

- This paper lacks experiments to show the necessity of using two-round decoding. If the independent chunk translation is already satifactory, then second decoding round using context is not necessary. Translation performance of the first and second decoding round should be compared to have a clear view of the usefulness of the second decoding round.
- Also the effect of using few-shot prompting with 5 examples is not revealed in this paper. Since the bidirectional mT5 already performs both the direct translation task by filling the mask at the end of a source sequence, and the text in-filling task by predicting the mask in the middle of a source sequence, it seems that zero-shot translation using the two round decoding strategy also works.

**Reproducibility:**

4: Could mostly reproduce the results, but there may be some variation because of sample variance or minor variations in their interpretation of the protocol or method.

**Reviewer Confidence:**

4: Quite sure. I tried to check the important points carefully. It's unlikely, though conceivable, that I missed something that should affect my ratings.

**Typos Grammar Style And Presentation Improvements:**

- L36: any references to support this statement?
- L64: mT5 is proposed in Xue et al instead of Lin et al
- Figure 3: It would be better if it shares the same form as Figure 4, i.e. texts instead of picture

---

> ### Author Rebuttal · Authors · 2023-08-28
>
> We thank the reviewer for their feedback including appreciation of our approach and evaluation.
>
> We first want to make the clarification regarding our approach:
>
> **Q: During the first decoding round, the authors split a source sentence into several chunks and performs independent translation of each chunk. (H_i -> M_i) In the second decoding round, each source chunk is concatenated with the neighbor chunks (left and right) as context (H_i-1 H_i H_i+1), and the corresponding translated context segment from round 1 is reused as target context (M_i-1 <mask> M_i+1). The model then performs a text in-filling task to output the translation correponds to the <mask> in the given context (H_i-1 H_i H_i+1: M_i-1 <mask> M_i+1 -> R_i). Finally the generated R_i are concatenated as the final translation.**
>
> The reviewer has aptly summarized the first stage of our two-stage approach. However, we'd like to clarity the approach in the second stage: In this stage, the input is H_i-1 H_i H_i+1 along with R_i-1 <mask> M_i+1 to predict R_i. So, the transformation can be represented as:
> (H_i-1 H_i H_i+1: R_i-1 <mask> M_i+1 -> R_i). This process is explained in Figure 1, Section 3.5, and in equations (2) and the subsequent equation.
>
> **Q: This paper lacks experiments to show the necessity of using two-round decoding. If the independent chunk translation is already satifactory, then second decoding round using context is not necessary. Translation performance of the first and second decoding round should be compared to have a clear view of the usefulness of the second decoding round.**
>
> As addressed in our response to Reviewer 2 (Q3), we have presented a comparison between the single-stage independent translation and DecoMT. The results consistently demonstrate DecoMT's superiority across all language pairs and directions.
>
> **Q: Also the effect of using few-shot prompting with 5 examples is not revealed in this paper. Since the bidirectional mT5 already performs both the direct translation task by filling the mask at the end of a source sequence, and the text in-filling task by predicting the mask in the middle of a source sequence, it seems that zero-shot translation using the two round decoding strategy also works.**
>
> As suggested, we undertook zero-shot translation experiments for select language pairs, specifically hin<->gujr, hin<->tel, and hin<->mal, given our computational constraints. We compared different approaches applied to mT5 including DecoMT, SAP and Standard Prompting. We found that all approaches yielded near-zero BLEU scores.  In most instances, the models merely copied the input as the output. We hypothesize that this is because in a zero-shot setting the model may not understand that it has to perform translation to the target language.
>
> **Q: L449, the authors mentioned that each language pair only have one single annotator. Does this cause randomness in the human evaluation results?**
>
> The human evaluations are useful to take a closer look and confirm what the automated metrics have shown. We concur that relying on a single annotator may introduce some variance in human evaluation. Nevertheless, to mitigate biases, we ensured that evaluation tasks were randomized, ensuring no system gets an undue advantage.
>
> We're also grateful for the additional references provided and will incorporate them to better contextualize our work. We will also fix the typos.

---

### Official Review · Reviewer_TKfW · 2023-08-03

**Soundness:** 4

**Excitement:**

3: Ambivalent: It has merits (e.g., it reports state-of-the-art results, the idea is nice), but there are key weaknesses (e.g., it describes incremental work), and it can significantly benefit from another round of revision. However, I won't object to accepting it if my co-reviewers champion it.

**Paper Topic And Main Contributions:**

This paper proposes a framework for related languages translation which decomposes the prompts into two stages, i.e. independent chunk translation and then a contextual translation. The related languages have the similar word order and linguistic characteristics so that it can be translated monotonically. To better use these features, the paper first separate the input sentence into a number of chunks and translate them independently, then by combining the contexts of the target translation, the framework performs an incremental/contextual translation. Experiments on different related languages show significant improvements on different evaluation metrics compared to baselines.

The main contribution of this paper is the method to decompose the prompts, realising a two-stage translation for related languages. However, this method is only suitable for similar family of languages.

**Questions For The Authors:**

1. I guess even for related languages, there still exists word order differences, how would this affect the translation quality?
2. Is it possible to use syntactic parser to split the input sentence into a number of chunks so that each chunk keeps a relatively complete semantic meaning?
3. How much is it improved by adding the contextual translation (2nd stage) compared to the independent translation (1st stage)?
4. If we remove the first stage, and only use the second stage, i.e. source chunks + previously translations + <mask>, what would the performance be?

**Reasons To Accept:**

The main contribution of this paper is the method to decompose the prompts, realising a two-stage translation for related languages.

**Reasons To Reject:**

From the experimental results we can see although the DecoMT performs better than SAP, it is not a huge margin, but the two-stage process makes the translation more complicated and not straightforward. Even authors claim that using parallel computing for independent chunk translation, it is not fair to compare the efficiency or speed in this way. It's fairer to compare under same computing resources.

**Reproducibility:**

4: Could mostly reproduce the results, but there may be some variation because of sample variance or minor variations in their interpretation of the protocol or method.

**Reviewer Confidence:**

4: Quite sure. I tried to check the important points carefully. It's unlikely, though conceivable, that I missed something that should affect my ratings.

---

> ### Author Rebuttal · Authors · 2023-08-28
>
> We thank the reviewer for the feedback. We address the concerns below:
>
> **Q: From the experimental results we can see although the DecoMT performs better than SAP, it is not a huge margin, but the two-stage process makes the translation more complicated and not straightforward.**
>
> Despite the two-stage design, DecoMT's methodology is intuitive and straightforward to implement. More importantly, our results reveals that DecoMT holds a statistically significant advantage over SAP across various criteria.
> We evaluated DecoMT against SAP using 7 language pairs, analyzed in both directions. This results in a comprehensive comparison across 14 directions, as tabulated in Table 1.
> In terms of the chrF++ metric, DecoMT considerably outperforms SAP across all 14 directions.
> When assessed using spBLEU, DecoMT exhibits superior performance in 12 directions, is comparable with SAP in 1 direction, and falls short in another.
> We've established the statistical significance of these findings using paired bootstrap sampling (Koehn, 2004) with a confidence level set at p<0.05.
> Our analyses, particularly depicted in Figure 5, further illustrate that DecoMT's superiority grows more pronounced as sentence lengths increase.
>
> **Q: Even authors claim that using parallel computing for independent chunk translation, it is not fair to compare the efficiency or speed in this way. It's fairer to compare under same computing resources.**
>
> To ensure a fair comparison, we've now profiled the codes using cprofile (https://docs.python.org/3/library/profile.html) during the inference phase, executed on an A40 48GB GPU. cprofile examines the time taken by various API calls. Our chosen task was translating from Marathi to Hindi using the initial batch of 5 examples from the FLORES test set, with the longest Marathi sample in the batch being 41 tokens long.
> 1. SAP Analysis: For the SAP system, due to the unpredictability of the expected target length, we do decoding at 1.5 times the maximum source length. This is based on our studies of lengths of examples from validation dataset. For example, for our given source batch, the reference Hindi translation encompasses 55 tokens for the Marathi sentence which is 41 tokens long.
> As the longest example is 41 tokens, we run inference for 41 * 1.5 = 61 steps.
> Below is a partial trace using cprofile.
> We see that for SAP, there are 61 calls to predict_output method. The method predict_output is responsible for running inference on the LLM. Each method takes 2.384 seconds. The inference of the batch takes 145.455 seconds.
>
> | ncalls | tottime | percall | cumtime | percall | filename:lineno(function)             |
> |--------|---------|---------|---------|---------|--------------------------------------|
> | 3/1    | 0.000   | 0.000   | 145.455 | 145.455 | {built-in method builtins.exec}      |
> | ...    |         |         |         |         |                                      |
> | 61     | 0.002   | 0.000   | 145.428 | 2.384   | few-shot-sap.py:163(predict_output)  |
> | ...    |         |         |         |         |                                      |
>
> 2. DecoMT Analysis: For Marathi-Hindi translations, we use a chunk size of 4.
> We first consider the independent translation stage. Breaking down the sentence lengths of the batch in tokens: 16, 30, 24, 41, and 28, we get respective chunk counts of 4, 8, 6, 11, and 7—aggregating to 36 chunks. Split into batches of 8, this leads to 5 API calls to predict_output.
> With the longest sentence in the batch having 41 tokens, the contextual translation stage demands 11 API calls to predict_output, cumulating to 16 calls.
> These 16 api calls in total amount to 96.883 seconds.
> While predict_output in DecoMT tends to take longer than in SAP (owing to DecoMT predicting multiple tokens as opposed to SAP's single-token approach), the overall fewer API calls render DecoMT more efficient.
>
> Below is a partial trace using cprofile.
> | ncalls | tottime | percall | cumtime | percall | filename:lineno(function)                    |
> |--------|---------|---------|---------|---------|----------------------------------------------|
> | 3/1    | 0.000   | 0.000   | 96.883  | 96.883  | {built-in method builtins.exec}              |
> | ...    |         |         |         |         |                                              |
> | 16     | 0.001   | 0.000   | 96.868  | 6.054   | few-shot-decomp-mt.py:199(predict_output)    |
> | ...    |         |         |         |         |                                              |
>
> We will incorporate this detailed analysis into the main paper for comprehensive understanding.
>
> **Q1: I guess even for related languages, there still exists word order differences, how would this affect the translation quality?**
>
> During the annotation of the 5 examples, annotators were instructed to maintain a monotonic alignment between the source and target. The annotators could reorder as well as modify the sentences to ensure this monotonicity. For the languages we have studied, the annotators were able to ensure monotonicity in the 5 examples.
> We believe that when word order differences arise during testing, they are typically localized. As a result, they can be effectively managed through intra-chunk reordering. Nonetheless, a deeper exploration of this question is part of our plan for future work.
>
> **Q2: Is it possible to use syntactic parser to split the input sentence into a number of chunks so that each chunk keeps a relatively complete semantic meaning?**
>
> At present, our method prioritizes a simpler strategy, employing a fixed chunk size for each source-target language combination. This approach has yielded satisfactory outcomes.  Leveraging syntactic parsers to achieve more semantically cohesive chunks is an interesting idea. However, most of the languages we consider in our work lack a good syntactic parser.
>
> **Q3: How much is it improved by adding the contextual translation (2nd stage) compared to the independent translation (1st stage)?**
>
> In a new set of experiments, we compared the single-stage independent translation to the two-stage DecoMT. The experiments show that the inclusion of contextual translation in the second stage of DecoMT significantly improves performance. Here are the results:
>
> | lang pair     	| spBLEU (Single Stage)  | spBLEU (DecoMT)  	| chrF++ (Single Stage)  | chrF++ (DecoMT)  	|
> |----------|------------------------|----------------------|------------------------|----------------------|
> | mar->hin | 17.0               	| 21.0             	| 39.6               	| 41.9             	|
> | hin->mar | 12.1               	| 13.9             	| 33.1               	| 35.6             	|
> | guj->hin | 21.0               	| 23.2             	| 41.8               	| 43.7             	|
> | hin->guj | 20.2               	| 22.0             	| 39.6               	| 41.1             	|
> | mal->hin | 13.3               	| 16.3             	| 34.7               	| 36.8             	|
> | hin->mal | 15.9               	| 18.7             	| 33.7               	| 37.0             	|
> | hin->tel | 16.7               	| 19.5             	| 36.3               	| 38.5             	|
> | tel->hin | 11.3               	| 17.8             	| 35.7               	| 38.6             	|
> | spa->por | 24.4               	| 26.5             	| 48.7               	| 50.0             	|
> | por->spa | 23.7               	| 26.3             	| 47.1               	| 48.9             	|
> | zsm->ind | 26.4               	| 29.6             	| 53.8               	| 55.9             	|
> | ind->zsm | 27.7               	| 28.2             	| 54.3               	| 54.5             	|
> | ukr->rus | 32.1               	| 34.4             	| 49.9               	| 51.5             	|
> | rus->ukr | 28.3               	| 31.0             	| 48.5               	| 49.9             	|
>
> These findings emphasize the importance of contextual translation in the DecoMT approach. We will include this ablation study in our paper.
>
> **Q4: If we remove the first stage, and only use the second stage, i.e. source chunks + previously translations + <mask>, what would the performance be?**
>
> The method you described is referred to as “Standard Prompting applied to mT5” in our paper (see lines 350-354). A direct comparison can be found in Table 1 under the column SP(mT5). Across all language pairs, SP(mT5) consistently scores lower in both spBLEU and chrF++ metrics when compared to DecoMT. This underscores the efficacy of the two-stage DecoMT approach.

---

### Official Review · Reviewer_F6Qp · 2023-08-04

**Soundness:** 2

**Excitement:**

2: Mediocre: This paper makes marginal contributions (vs non-contemporaneous work), so I would rather not see it in the conference.

**Paper Topic And Main Contributions:**

The paper proposes a prompt scheme for MT of closely related languages. The method is in two stages:
  - In a first stage, the input is chunked and each chunk is translated separately, using a few shots of monotonic-translated examples
  - In a second stage, which the authors refer to as "contextual translation", the translated chunks are retranslated iteratively, with the previous output supplied in the target as context.

Decomposed prompting is inspired by Khot et at 2023. The innovation is claimed to be in the second stage, the "contextual translation"
The motivation for this scheme is that for closely related languages, the first stage encourages monotonic translation (through the few shots presented in the prompt). This is more efficient since these languages tend to be low-resource.

The experiments use mT5

The paper is thin on innovation, and the context is not enough for a full-length EMNLP paper

**Questions For The Authors:**

- How is the input chunked to ensure that the chunks can be translated monotonically?

**Reasons To Accept:**

- Evaluation on multiple language pairs from different language families is included

- The paper is well written and the method is well-explained



**Reasons To Reject:**

- The innovation is not enough for an EMNLP full-length paper

- The method has an inefficiency, since it required multiple inference calls (as many as the number of chunks) in the contextual translation stage, which cannot be parallelized since the outputs are used successively

**Reproducibility:**

3: Could reproduce the results with some difficulty. The settings of parameters are underspecified or subjectively determined; the training/evaluation data are not widely available.

**Reviewer Confidence:**

4: Quite sure. I tried to check the important points carefully. It's unlikely, though conceivable, that I missed something that should affect my ratings.

---

> ### Author Rebuttal · Authors · 2023-08-28
>
> Thank you for your feedback, particularly for appreciating our evaluation on multiple language pairs from different language families. We address your concerns below.
>
> **Q1. The innovation is not enough for an EMNLP full-length paper**
>
> We've introduced Decomposed Prompting for Machine Translation (DecoMT).
> * Incremental Translation: Unlike conventional methods using autoregressive prompting of LLMs like BLOOM and XGLM that process entire sentences, DecoMT translates word chunks incrementally. This approach yields an 8 chrF++ improvement over BLOOM across language pairs.
> * Monotonic Alignment: Current few-shot prompting methods neglect intrinsic traits of related languages. DecoMT addresses this by focusing on inherent monotonic alignment, improving translation accuracy.
> * Focus on Less-explored Language Pairs: We've experimented with less-explored language pairs such as hindi<->malayalam, indonesian<->malay, and ukrainian<->russian.
> * Bidirectional Context: Distinct from Sequential Autoregressive Prompting (SAP) from Patel et al. (2023), DecoMT uses bidirectional context in translation, as referenced in lines 139-143.
>
> Empirically, our evaluations cover a wide range of language pairs. Through both automatic and human evaluation, we consistently outperform established baselines. Additionally, our analysis reveals that DecoMT achieves higher chrF++ scores than SAP as source sentence length increases.
>
> **Q2: The method has an inefficiency,  since it required multiple inference calls…**
>
> We acknowledge DecoMT's inefficiency compared to standard autoregressive prompting based approaches, as noted in lines 557-562 in the Limitations section. However, it's pertinent to note that DecoMT still stands as more efficient than SAP, as detailed in lines 524-534.
> Most LLMs have limited representation of the majority of non-English languages, hence it is imperative to introduce a specific inductive bias. In our case, the inductive bias is that of monotonicity between related languages. The inductive bias we chose significantly improves translation quality, which we believe justifies the increased cost.
>
> **Q3: How is the input chunked to ensure that the chunks can be translated monotonically?**
>
> During the annotation of the 5 examples, annotators were instructed to maintain a monotonic alignment between the source and target. Take, for instance, the translation from Marathi to Hindi. In DecoMT's first stage, there were 33 chunks spanning the 5 examples in the prompt template. We expect that, given these 33 in-context examples, the model learns the task of monotonic translation. In DecoMT's second stage, the chunks are longer, resulting in fewer of them. Continuing with the Marathi to Hindi translation example, there were 19 chunks in the prompt template. We observed that given these in-context examples, the model adeptly learns infilling. In this stage, the model can leverage a broader local context from translations of neighboring chunks as well as the associated source chunks to ensure accurate translation and refine the first stage's translation. Consequently, our straightforward chunking approach effectively ensures that translations proceed in a monotonic manner.
>
> **Q4: Reproducibility**
>
> Upon acceptance, we will release the code and model outputs.

---

### Meta-Review · Area_Chair_8N9S · 2023-09-18

**Recommendation:** 4

**Metareview:**

The paper proposes an interesting two-pass method for promting LLMs for translation between related languages.

The proposed method yields better performance in terms of automatic evaluation scores than previous prompting methods over several language pairs from different families.

The method may be an inspiration for future work on translation between non-English or low-resourced languages using LLMs.

The revisions arised during authors' response should be applied in the final version.

---

### Decision · Program_Chairs · 2023-10-07

**Decision:**

Accept-Main

**Comment:**

The paper proposes an interesting two-pass method for promting LLMs for translation between related languages.

The proposed method yields better performance in terms of automatic evaluation scores than previous prompting methods over several language pairs from different families.

The method may be an inspiration for future work on translation between non-English or low-resourced languages using LLMs.

The revisions arised during authors' response should be applied in the final version.